# An exploration of multi-level factors affecting routine linkage to HIV care in Zambia's PEPFAR-supported treatment program in the treat all era

Jenala Chipungu[¤a]*, Helene Smith[¤a], Chanda Mwamba, Mwiza Haambokoma, Anjali Sharma, Theodora Savory, Maurice Musheke, Jake Pry, Carolyn Bolton[¤b], Izukanji Sikazwe, Michael E. Herce[¤c]

Research Department, Social and Behavioral Science Unit, Centre for Infectious Disease Research in Zambia (CIDRZ), Lusaka, Zambia

¤a Current address: School of Population Health, University of New South Wales, Sydney, Australia
¤b Current address: Division of Infectious Diseases, Department of Medicine, University of Alabama at Birmingham, Alabama, United States of America
¤c Current address: Institute for Global Health & Infectious Diseases, University of North Carolina, Chapel Hill, North Carolina, United States of America
* Jenala.Chipungu@cidrz.org

## Abstract

Multiple steps from HIV diagnosis to treatment initiation and confirmed engagement with the health system are required for people living with HIV to establish full linkage to care in the modern treat all era. We undertook a qualitative study to gain an in-depth understanding of the impeding and enabling factors at each step of this linkage pathway. In-depth interviews were conducted with fifty-eight people living with HIV recruited from ten routine HIV care settings supported by the U.S. President's Emergency Plan for AIDS Relief (PEPFAR) in Lusaka, Zambia. Using a semi-structured interview guide informed by an established conceptual framework for linkage to care, questions explored the reasons behind late, missed, and early linkage into HIV treatment, as well as factors influencing the decision to silently transfer to a different clinic after an HIV diagnosis. We identified previously established and intersecting barriers of internal and external HIV-related stigma, concerns about ART side effects, substance use, uncertainties for the future, and a perceived lack of partner and social support that impeded linkage to care at every step of the linkage pathway. However, we also uncovered newer themes specific to the current test and treat era related to the rapidity of ART initiation and insufficient patient-centered post-test counseling that appeared to exacerbate these well-known barriers, including callous health workers and limited time to process a new HIV diagnosis before treatment. Long travel distance to the clinic where they were diagnosed was the most common reason for silently transferring to another clinic for treatment. On the other hand, individual resilience, quality counseling, patient-centered health workers, and a supportive and empathetic social network mitigated these barriers. These findings highlight potential areas for strengthening linkage to care and addressing early treatment interruption and silent transfer in the test and treat era in Zambia.

**Data Availability Statement:** Data was primarily qualitative data comprising of 58 audio recordings and corresponding transcripts. Additionally, each participant completed a socio-demographic form that was linked to their transcripts. Transcripts have not been shared in a data depository given the sensitivity of the information shared and the risk of misinterpretation by others lacking sufficient information on the context of the data shared. However, selected relevant excerpts have been shared in this publication to support the findings. However, those requiring detailed excerpts related to the objective of the study can reach out to the following: Jenala Chipungu Head Social and Behavioural Science Research Department Centre for Infectious Disease Research in Zambia (CIDRZ) Jenala.Chipungu@cidrz.org Professor Victor Chalwe Acting Director/Chief Executive Officer National Health Research Authority NATIONAL HEALTH RESEARCH AUTHORITY Lot No. 18961/M, off Kasama Road, Chalala, P.O. Box 30075, LUSAKA Tell: +260211 250309/0777409076 | Email: zhrasec@nhra.org.zm | www.hra.org.zm

**Funding:** This work was supported by the PEPFAR, through funding from the US Centers for Disease Control and Prevention (CDC) Global AIDS Program in Zambia to the Centre for Infectious Disease Research in Zambia (CIDRZ), (Cooperative Agreement NU2GGH001920 to IS). The funders had no role in study design, data collection and analysis, decision to publish, or preparation of the manuscript.

**Competing interests:** The authors have declared that no competing interests exist.

# Introduction

In December 2016, Zambia adopted "test and treat" nationally, in which every person living with HIV (PLHIV) is eligible to start ART, regardless of CD4 count or WHO clinical stage [1]. Under these parameters, the country now faces the difficult challenge to meet the "95-95-95" goals set by UNAIDS to ensure that by 2030, ≥95% of PLHIV know their status, ≥95% of PLHIV are on treatment and, of those, ≥95% are virologically suppressed [2].

Whether ambitious 95-95-95 targets can be achieved remains uncertain. The Zambia Population-based HIV Impact Assessment (ZAMPHIA) conducted in 2021 estimated that 88.7% of adults (15–59 years old) knew their HIV status, of whom, 98.0% reported current ART use and 96.3% of ART-treated PLHIV were virologically suppressed [3]. While the most recent ZAMPHIA data reveal remarkable progress for ART uptake and virological suppression, gaps remain in HIV case finding, with important implications for sustaining high linkage to care. Results from the PopART trial—a three-arm community randomized trial in Zambia and South Africa evaluating the impact of universal test and treat (UTT) on HIV incidence—found that while acceptance of HIV testing in the community was high, linkage to care and ART initiation took longer than expected, with only 53% linked to care 12 months after their initial diagnosis [4].

Many studies on factors contributing to linkage to care describe varying individual, social, health system, and structural barriers faced by patients in this process [5–7]. In low- and middle-income countries (LMIC) globally, commonly cited barriers include, among others, navigating complex care registration systems, the availability of health workers to start clients on ART and client readiness to initiate ART, inadequate support from family members and the health system, the use of traditional medicines, concomitant global calamities such as the COVID-19 pandemic, and concerns surrounding confidentiality [8–10]. Importantly, stigma and discrimination have also been reported as serious hinderances to linkage to HIV care [11]. In Zambia, the spatial organization of HIV services within clinics, where the ART department is a demarcated stand-alone building or room visible to the public, and overcrowding are prominent barriers, which contribute directly to the fear of being seen and stigmatized by others [8, 12]. While foundational to our understanding of linkage to care, these studies generally preceded the universal test and treat era and conceptualized linkage as a one-step referral process in which treatment initiation directly follows HIV testing and counseling.

A closer examination of linkage to care in the context of universal test and treat, however, reveals a potentially more nuanced process requiring a series of key steps in the pathway to fully and durably achieve linkage to care. The aim of this study was to obtain qualitative data to provide explanatory power and understanding surrounding the context, factors, and processes enabling or impeding 'full' linkage to care in the modern treat all era, with a special focus on routine clinical practice settings in high-burden LMICs where HIV services are normally delivered.

# Methods

## Study design

**Study overview.** The qualitative study was part of a larger mixed methods study, "Linkage to Care and Treatment in the Era of Test and Start: A Mixed Methods Evaluation in Zambia (short title: *LINC*)," which aimed to estimate the extent to which newly diagnosed PLHIV are linked into care and treatment within 6 months of HIV diagnosis at public health facilities in Lusaka, Zambia. Participants were recruited between the 2nd January, 2019 and 29th March, 2019. Study procedures and findings are reported here according to established criteria for qualitative research [13].

**Theoretical framework.** For this qualitative study, we conceptualized linkage to care according to a previously published framework [14]. The framework describes the pathway, or series of discrete steps, that a newly HIV-diagnosed person must negotiate to demonstrate "full linkage," or sufficient commitment and support to continue HIV care [14]. These steps include: 1) the initial experience of HIV testing, 2) post-test counseling, 3) the transition of the client from the HIV testing service venue to the treatment venue, 4) the clinical evaluation, 5) initial collection of ART drugs, 6) psychosocial support and follow-up, and 7) return contact with the health system for a first ART refill and to confirm HIV care engagement. Under this framework, a client achieves full linkage when they complete this final step. The qualitative component of the *LINC* study, presented here, applied a phenomenological perspective to understand the reasons why patients establish, or fail to establish, full linkage to care.

*Study Setting.* The study was conducted in 10 government health facilities supported by the Centre for Infectious Disease Research in Zambia (CIDRZ), with funding from the U.S. President's Emergency Plan for AIDS Relief (PEPFAR), in Lusaka Province. Lusaka Province, which includes the country's capital city and a mix of urban, peri-urban, and rural sites, has the highest HIV prevalence in the country (14.4%) with 87.9% of that population virologically suppressed [3].

The facilities were purposively sampled to capture the demographic diversity of urbanicity within the province, as well as high and low rates of linkage to care based on routine PEPFAR performance indicators. At all study sites, healthcare staff follow newly diagnosed clients using two paper-based registers: a HIV Testing Service (HTS) and an ART Linkage register. Clients agreeing to initiate ART are enrolled in the national ART program where their medical records are maintained using a mix of electronic medical record (EMR, known as SmartCare) and paper-based patient files.

*Participant Selection.* At each facility, clinic staff reviewed both HTS and ART Linkage registers to identify potential participants for the qualitative study who had tested HIV positive in the last 6 months. To be study eligible, participants had to have a discernable "linkage status" in the routine record that generally aligned with one of the different steps along the linkage to care pathway [Table 1] [14]. For each participant, including those in the silent transfer category, study staff working with clinic staff carried out a two-step process to confirm linkage status category. In the first step, potential participants identified from paper-based registers had their EMR record reviewed for documented instances of prior clinic encounters or ART collection at the study sites or any other facility in Lusaka. In the second step, potential

**Table 1. Categories of linkage status for study participants.**

| Linkage Status | Definition |
|---|---|
| No enrolment (NE) | Never enrolled in the national ART/ HIV treatment program after HIV diagnosis |
| No ART Pick Up (NA) | Enrolled but never collected their first supply of ART |
| No First Follow-up (NF) | Enrolled and collected their first ART prescription, but never returned for a follow-up encounter with the health system |
| Late Linkage (LL) | Completed the linkage to care pathway, including a first follow up visit, but did so at least 90 days from their initial HIV diagnosis |
| Early Successful Linkage (EL) | Completed the full linkage to care pathway, including a first follow up visit, within 90 days of their initial HIV diagnosis |
| *Silent Transfer (ST) | Transferred to another clinic to continue their ART treatment without notifying the clinic where they first received their HIV diagnosis and/or established care |

*Assessed through staff review of the routine record and confirmation with potential participants during study recruitment.

participants had their linkage status confirmed through direct questioning at the time of study recruitment.

Finally, in order to be eligible, participants needed to be ≥16 years old, able to converse freely in a study language (English, Nyanja or Bemba), able to provide written informed consent and willing to participate in an in-depth interview.

**Sampling.** Participant sampling was purposive and followed the schema below (see *Sample Size*). Trained lay healthcare workers employed by each clinic to trace clients with missed appointments and support HIV care engagement, contacted potential participants identified from HTS and ART Linkage registers. Using a phone script, lay healthcare workers briefed the participants on the study and referred those interested to a research assistant. The research assistant then scheduled a screening and enrollment visit with the participant at a time and location of their choosing. During the visit, the research assistant reviewed the aims of the study with the participant, including the name of the Principal Investigator and reasons for doing the research, using an approved script and confirmed eligibility for the study (including confirming that their linkage status matched what had been ascertained from the routine record). If eligible, the participant completed informed consent procedures and a simple demographics form before proceeding to an in-depth interview (IDI). At the end of the interview, the research assistant in consultation with the study team (JC & CM) established whether the participant needed to be referred for additional services (including being re-connected to care, psychosocial counselling, etc.) and provided support as needed.

**Sample size.** We aimed to conduct approximately 60 IDIs across the 10 study sites, with about 10 HIV-positive clients enrolled from each of the following six linkage status categories (Table 1). These comprised PLHIV who: i) failed to enroll in the national ART/ HIV treatment program (NE); ii) enrolled but did not have a baseline clinical evaluation/ first ART pick up (NA); iii.) enrolled and collected their first ART prescription, but did not return for a first follow-up visit (NF); iv.) completed the linkage to care cascade but did so late (LL); v.) completed the linkage to care cascade but did so as a silent transfer (ST); and vi) completed the full linkage to care pathway, including a first follow up visit, within 90 days of their initial HIV diagnosis (EL).

**Data collection.** The co-investigators named above and the research assistants working with them collected study data at the study sites or a convenient location of the participant's choosing in the community. All data were collected in English, Nyanja, or Bemba based on participant preference. Research assistants administered a brief demographics form. Subsequently, one-time IDIs were conducted face-to-face in the absence of other persons apart from the participant and researchers. Experienced female qualitative researchers, JC and CM oversaw the data collection and reviewed recordings of each in-depth interview conducted by the research assistants for quality control. At the time of the study, JC and CM were Masters level full-time behavioral scientists working at CIDRZ. They both expressed interest in the research topic and underwent a 5-day training session on the study protocol, including all procedures and IDI guide administration, by the study Principal Investigator, MEH, and Senior CIDRZ Technical Advisor, AS.

Demographics Form: Using a structured case reporting form, research assistants collected demographic data on participant HIV testing entry-point, gender, marital status, education level, household income, source of livelihood, and approximate travel distance between home and clinic.

In-Depth Interview Guide: IDIs used a semi-structured interview guide developed to explore empirically supported factors and other issues hypothesized to influence linkage to care according to our theoretical framework, including stigma, psychosocial support, social networks, gender inequality, male partner involvement, health service quality, healthcare

worker attitudes, and financial barriers, among others. The IDI guide was pilot tested prior to field data collection.

During the in-depth interview (IDI), participants were asked by the researchers to describe: their reasons for testing and their experiences with the testing and counseling process; their experiences being linked to HIV care and, if they didn't link to care, the individual, social and structural drivers for this decision; relationships that may have influenced their knowledge and general attitudes towards HIV, linking to care, and initiating ART; and, if applicable, their reasons for silently transferring to a new clinic(s). The IDI guides were applied flexibly, and participants' own language styles were adapted and incorporated into additional probes as needed. Probing and follow-up questions were used to clarify responses and encourage elaboration. Each interview lasted between 60 and 90 minutes. Non-verbal language, gestures and participants demeanor were noted down in a summary form that each research assistant had to complete after conducting the interviews. All IDIs were audio-recorded for accuracy. No repeat interviews were conducted.

## Data analysis

Each interview was transcribed verbatim and translated into English by a separate team of 3 transcribers. JC conducted quality control (QC) spot checks by reviewing portions of transcripts and associated audio files to ensure accuracy in translations and transcriptions. Transcripts were not returned to participants for comment and participants did not provide feedback on the findings. Upon passing QC, interview transcripts were uploaded into NVIVO (QSR International, Australia) for coding.

A thematic framework stratified by linkage status was developed for data analysis using inductive reasoning based on emerging themes capturing factors that facilitate and hinder linkage to care [8]. Three independent coders (JC, CM & MH) coded two transcripts each to refine and validate the codes, followed by another three transcripts each to achieve consistency in coding. The remaining transcripts were then coded independently. Data was indexed by themes as well as cases including type of respondent by age and gender. Indexed data was categorized by linkage status (Table 1) and put into matrixes corresponding to reasons for testing and facilitators and barriers to linkage to care, identifying connections between themes, patterns, and inherent evidence in extracted quotes.

The codebook captured codes representing data on *knowledge* held on HIV before and after testing, followed by their *decision making* for and *experience* (positive and negative) with HIV testing. Additional codes and sub-codes were elaborated on feelings, thoughts and reactions that participants experienced after receiving their HIV test result. *Linkage to care* was a main code that described barriers and facilitators to treatment initiation and care engagement, interactions with health care staff through the triage process. It also captured participant decision regarding accepting or refusing treatment as well as influencing factors affecting their decisions and experience taking their anti-retroviral (ARVs) drugs. Another code was *psychosocial support* with multiple sub codes on coping strategies for accepting their HIV status, support such as mental, social, emotional, or spiritual received by their social networks, and their experience with disclosure of their HIV status and stigma. Finally, the code *patient treatment status* encompassed data on specific reasons for dis-engagement from care including reasons for silent transfer.

Coders followed Braun and Clarke's six step analysis process including familiarization, code generation, development of themes, refining and naming of themes and finally a written interpretation of the themes [15]. Regular meetings at each step in the analysis process were held to discuss emerging codes, their meanings and data saturation. Final discussions included

an agreement between coders on the themes and their interpretations and the output was a report outlining the reasons for dropouts in each of the linkage status categories with supporting quotes.

### Ethics statement

Written informed consent was obtained from all study participants. Where a participant was illiterate, an impartial witness selected by the participant was asked to witness the reading of the information sheet and provide written consent confirming that the participants had freely consented. The study was approved by the University of Zambia Biomedical Research Ethics Committee (#011-12-17), the U.S. Centers for Disease Control and Prevention (#2018–381), and the University of North Carolina at Chapel Hill Institutional Review Board (#18–0854).

## Results

Fifty-eight participants living with HIV from 10 clinics were identified, enrolled, and completed an interview across linkage categories as follows: No Enrolment (n = 8), No ART Pickup (n = 8), No First Follow-up (n = 11), Late Linkage (n = 10), Early Linkage (n = 10) and Silent Transfer (n = 11). A plurality of women linked to care, but did so late or without a first follow up visit, while male participants appeared to drop off the linkage pathway earlier, and more commonly did not enroll in the national HIV treatment program at all or never presented for a first ART collection (Table 2). As expected, most participants received HTS at the outpatient department (OPD, 44%) and the voluntary counselling and testing (VCT) corner in the clinic (32%). Linkage categories were fairly evenly distributed for both these testing venues, with slightly more participants having No First Follow up visit in these entry HIV testing locations. Among married participants, the most common linkage status was silent transfer (29%).

### Reasons for HIV testing

Ninety six percent of participants (56 of 58) provided narrations describing how they came to know their HIV status. Of these, 57% (32 of 56) reported receiving provider-initiated testing and counselling (PITC) when presenting for medical attention. Of PITC recipients, 50% (17 of 32) presented for an illness they did not relate to HIV (e.g., hypertension, physical injury or trauma), and 25% (8 of 32) were offered PITC in the course of seeking other health services, including routine antenatal care, family planning, voluntary male medical circumcision, or pediatric services for their under-5-year-old child. Five of 32 (16%) participants accompanied their partners to the clinic and were offered PITC when their partners tested HIV positive (Fig 1).

Several participants in the PITC group mentioned not being mentally prepared to learn their HIV status at the time of testing as they sought medical attention for other reasons.

> "I had no intentions of testing when I went there [to the clinic], but while there I was asked to test before seeing the doctor and receiving treatment. I was told that was the procedure, [asked] is it mandatory?—[was told] 'yes'—and that 'you cannot be treated if you have not gone through VCT. So, after being tested that is when you can proceed to see the doctor to examine you on other things.' But I had no intentions to go and test for VCT."–Single man, 33 years old, No First Follow-up

Aside from those who received PITC, 39% (22 of 56) of participants reported learning their status by presenting for VCT, and 4% (2 of 56) reported being tested through testing initiatives in the community. Among the VCT group, 45% (10 of 22) were concerned that they had acquired HIV because of perceived HIV risk from having unprotected sex, multiple sexual

**Table 2. Linkage category distribution by study participant characteristic (N = 58, row percentages reported).**

| Characteristics | NO ENROLMENT | NO ART PICK-UP | NO FIRST FOLLOW UP | LATE LINKAGE | EARLY LINKAGE | SILENT TRANSFER | TOTAL |
|---|---|---|---|---|---|---|---|
| | (n = 8) | (n = 8) | (n = 11) | (n = 10) | (n = 10) | (n = 11) | (N = 58) [a] |
| | n (%) | n (%) | n (%) | n (%) | n (%) | n (%) | N (%) |
| **Age in years** | | | | | | | |
| Median Age, years (interquartile range, IQR) | 30.5 (27, 34.5) | 27.5 (24, 34.5) | 30.5 (26, 40) | 30 (25, 31) | 26 (24, 32) | 34 (28, 39) | 35 (25.5, 34) |
| **Gender** | | | | | | | |
| Female | 3 (7.9) | 3 (7.9) | 9 (23.7) | 8 (21.1) | 7 (18.4) | 8 (21.1) | 38 (100) |
| Male | 5 (25.0) | 5 (25.0) | 2 (10.0) | 2 (10.0) | 3 (15.0) | 3 (15.0) | 20 (100) |
| **HIV entry test[b]** | | | | | | | |
| VCT | 1 (5.3) | 3 (15.8) | 4 (21.1) | 4 (21.1) | 3 (15.8) | 4 (21.1) | 19 (100) |
| MCH | 1 (20.0) | 0 | 0 | 2 (40.0) | 0 | 2 (40.0) | 5 (100) |
| OPD | 3 (11.5) | 4 (15.4) | 7 (26.9) | 4 (15.4) | 4 (15.4) | 4 (15.4) | 26 (100) |
| TB | 0 | 1 (100) | 0 | 0 | 0 | 0 | 1 (100) |
| Community Testing | 1 (50.0) | 0 | 0 | 0 | 1 (50.0) | 0 | 2 (100) |
| Other | 2 (40.0) | 0 | 0 | 0 | 2 (40.0) | 1 (20.0) | 5 (100) |
| **Attended School** | | | | | | | |
| Yes | 8 (14.5) | 8 (14.5) | 11 (20) | 10 (18.2) | 9 (16.4) | 9 (16.4) | 55 (100) |
| No | 0 | 0 | 0 | 0 | 1 (33.3) | 2 (66.7) | 3 (100) |
| **Highest Education level (year)** | | | | | | | |
| Primary (1–7) | 0 | 1 (7.1) | 3 (21.4) | 4 (28.6) | 2 (14.3) | 4 (28.6) | 14 (100) |
| Basic (8–9) | 5 (20.8) | 3 (12.5) | 6 (25.0) | 4 (16.7) | 4 (16.7) | 2 (8.3) | 24 (100) |
| Secondary (10–12) | 3 (25.0) | 3 (25.0) | 1 (8.3) | 1 (8.3) | 2 (16.7) | 2 (16.7) | 12 (100) |
| Tertiary | 0 | 1 (20.0) | 1 (20.0) | 1 (20.0) | 1 (20.0) | 1 (20.0) | 5 (100) |
| **Marital status** | | | | | | | |
| Married | 3 (12.5) | 3 (12.5) | 2 (8.3) | 6 (25.0) | 3 (12.5) | 7 (29.2) | 24 (100) |
| Single | 4 (17.4) | 4 (17.4) | 5 (21.7) | 4 (17.4) | 5 (21.7) | 1 (4.3) | 23 (100) |
| Divorced | 0 | 0 | 1 (25.0) | 0 | 1 (25.0) | 2 (50.0) | 4 (100) |
| Widowed/Separated | 0 | 1 (20.0) | 3 (60.0) | 0 | 0 | 1 (20.0) | 5 (100) |
| Other | 1 (50.0) | 0 | 0 | 0 | 1 (50.0) | 0 | 2 (100) |
| **Source of Livelihood** | | | | | | | |
| Formally Employed | 1 (12.5) | 1 (12.5) | 2 (25.0) | 0 | 2 (25.0) | 2 (25.0) | 8 (100) |
| Unemployed | 1 (10.0) | 3 (30.0) | 2 (20.0) | 2 (20.0) | 1 (10.0) | 1 (10.0) | 10 (100) |
| Student | 0 | 1 (33.3) | 2 (66.7) | 0 | 0 | 0 | 3 (100) |
| Small vendor | 0 | 0 | 1 (9.1) | 4 (36.4) | 2 (18.2) | 4 (36.4) | 11 (100) |
| Seasonal/ Farming | 5 (31.3) | 2 (12.5) | 3 (18.8) | 1 (6.3) | 3 (18.8) | 2 (12.5) | 16 (100) |
| Caretaker | 0 | 0 | 1 (14.3) | 2 (28.6) | 2 (28.6) | 2 (28.6) | 7 (100) |
| Other | 1 (50.0) | 0 | 0 | 1 (50.0) | 0 | 0 | 2 (100) |
| Refused to Answer | 0 | 1 (100.0) | 0 | 0 | 0 | 0 | 1 (100) |
| **Travel Distance to Nearest Clinic[c]** | | | | | | | |
| <30 min | 3 (21.4) | 3 (21.4) | 1 (7.1) | 2 (14.3) | 2 (14.3) | 3 (21.4) | 14 (100) |
| 3 0min-1 hour | 4 (13.8) | 3 (10.3) | 7 (24.1) | 5 (17.2) | 4 (13.8) | 5 (17.2) | 29 (100) |
| 1 hour-2 hours | 1 (7.7) | 2 (15.4) | 3 (23.1) | 1 (7.7) | 4 (30.8) | 2 (15.4) | 13 (100) |
| 2 hours-4 hours | 0 | 0 | 0 | 1 (100.0) | 0 | 0 | 1 (100) |
| Do not know | 0 | 0 | 0 | 1 (100.0) | 0 | 0 | 1 (100) |

[a] Numbers not summing to 58 reflect missing data.

[b] VCT = voluntary, counselling and testing (VCT); OPD = out-patient department; MCH = mother child health (MCH); TB = tuberculosis

[c] Clients were asked for travel distance to the government clinic nearest to their home

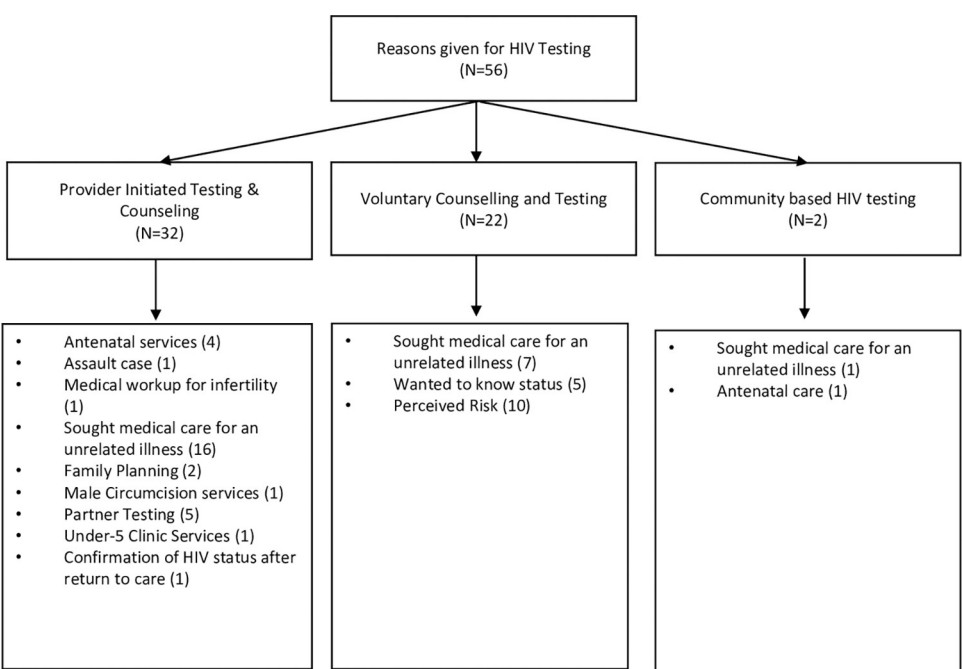

**Fig 1. Reasons for undergoing HIV testing.**

partners, or suspecting that their current sexual partner was living with HIV. Another 32% (7 of 22) experienced symptoms that they believed were suggestive of HIV (e.g., wasting, cough, and herpes zoster), while 23% (5 of 22) were simply interested in knowing their HIV status. For the two participants who tested in the community, one was encouraged to test because she was pregnant and the other as part of a general health check.

> "What caused me to have a test? _Yes. [Silence] [Clears throat] my body _aaeehh [agreeing sound]. . . I started to feel different. As I would walk, I would feel I was losing strength. . . I was being laughed at that I had lost weight. So that's how I bothered my young sibling for transport money so that I could test and know my status."–Married woman, 43 years old, No First Follow-up

## No enrolment

Eight participants did not enroll into HIV care after their HIV diagnosis. Although this occurred more commonly in men, both men and women mentioned several reasons for not wanting to enroll, including a lack of information on what to expect with the process of enrolment into the national HIV treatment program, having to work during normal clinic operating hours, denial about being HIV positive, fear of medication side effects, and avoidance of being seen by others at the clinic as a result of perceived stigma. One participant relayed confusion about what to do following his HIV diagnosis due to a lack of information.

> "They introduced me [to ART] but they didn't even tell me I am supposed to come start [ART] . . . I was waiting to be called back. Because I was not given any document to say am supposed to get back. I was requested to leave the phone number, from that time I have not received any call, I thought since they have my number I was expecting to be called. Since that time no phone came through, so I relaxed."–Married man (unknown age)

Fear of being stigmatized and fear of medication side effects featured prominently as reasons to not enroll. Participants recounted stories circulating in their communities about ART medication coming in pills that are too big to swallow or having bad side effects, including causing hallucinations or mental illness, weight gain, skin discoloration (Box 1), swelling, or body sores. Others struggled to come to terms with the idea of having to take ART for the rest of their lives. These fears, coupled with confusion about their HIV-related prognosis and concerns about the future of their intimate relationships, deterred these participants from enrolling in HIV care.

### Box 1. Narrative 1: Excerpts showing multiple psychosocial factors affecting decision to not enroll in HIV care (Single man, aged 32 years)

"My life has been full of question marks because I don't know where I'm going to start. From [the idea of] telling my partner about my status and again from the time I got tested, I have never sat and talked to anyone about it [my HIV]. Let's just say I have not opened up to anyone. The only person I can open up to is my girlfriend because she is the one close to me now. I keep asking myself how she is going to take it because we don't use condoms when we have sex. So, I don't know whether she will leave or stay with me when I tell her about my status. This is what has been troubling me.""I was scared of being put on treatment because those pills are too big, it would have been better if these pills were made much smaller. And then all those chemicals that make people dream of ghosts, gain weight, and get dark skin should be removed because how can medicine that is meant to make us better do all this to us?"

### No ART pick-Up

Eight participants did not pick up their ART medication after enrolling in the national HIV treatment program. Reasons for a missed first ART pick up were similar to those provided for not enrolling in HIV care. Among female participants (n = 3, 8% of women), these included failure to accept their HIV status and a bad experience with post-test counselling. In addition, women expressed a preference for injectable medications over pills. One woman who reported poor counselling said she felt discriminated against by the counsellor who spoke negatively of PLHIV during her counselling session. These feelings of discrimination and experiences of externalized stigma were compounded by feelings of depression, fear of side effects, and concerns about being able to swallow the tablets due to their size.

> "A counselor said bad things about people with HIV and I was hurt by the bad language . . . So I left."–Singe woman, 22 years old

The men (n = 5, 25% of men) who did not pick up their ART reported not feeling ready to start treatment and revealed underlying reasons of not feeling sick, not accepting their HIV status, and concerns about causing marital problems (Fig 2). They also described fear of disclosing their HIV status to others and concerns about being able to take ART consistently while dealing with unaddressed co-morbid hazardous alcohol use. One participant who disclosed hazardous alcohol use recounted feeling particularly upset about pressure to test during PITC, poor counselling about the rapidity of ART initiation and the need for treatment.

## CASE REPORT

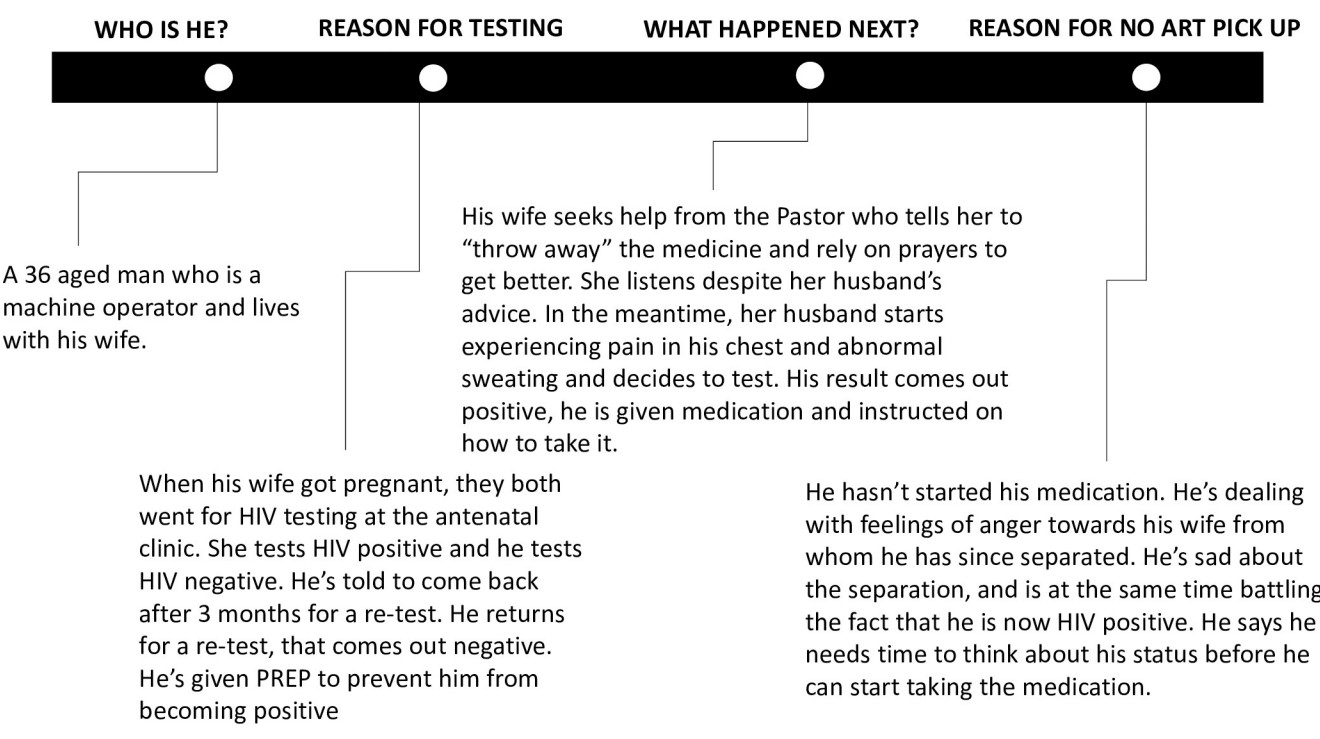

**WHO IS HE?**

A 36 aged man who is a machine operator and lives with his wife.

**REASON FOR TESTING**

When his wife got pregnant, they both went for HIV testing at the antenatal clinic. She tests HIV positive and he tests HIV negative. He's told to come back after 3 months for a re-test. He returns for a re-test, that comes out negative. He's given PREP to prevent him from becoming positive

**WHAT HAPPENED NEXT?**

His wife seeks help from the Pastor who tells her to "throw away" the medicine and rely on prayers to get better. She listens despite her husband's advice. In the meantime, her husband starts experiencing pain in his chest and abnormal sweating and decides to test. His result comes out positive, he is given medication and instructed on how to take it.

**REASON FOR NO ART PICK UP**

He hasn't started his medication. He's dealing with feelings of anger towards his wife from whom he has since separated. He's sad about the separation, and is at the same time battling the fact that he is now HIV positive. He says he needs time to think about his status before he can start taking the medication.

**Fig 2. No ART pick-up case report summary (married man, 36 years old).**

### No first follow-up

Eleven participants did not return for a subsequent ART clinic visit or second ART drug collection after receiving their initial prescription (typically 2 weeks' worth during the study period). The reasons given for not returning included HIV status denial, fear of side effects, feeling healthy and well, fear of losing employment because of perceived workplace discrimination and long travel distances to the clinic. HIV status denial featured prominently among participants who described initially not being prepared to test. For example, one participant reported that he was not going to start his ART because he was not convinced that he was HIV-positive since he did not feel sick or have any symptoms suggestive of HIV. Echoing other participants' concerns, those who did not return for a first follow-up cited their own, or their friends' and family members' experiences with, medication side effects. Participants reported feeling dizzy, having bad dreams, noticing skin color changes, or simply feeling unwell after taking medication. One male participant decided to discontinue ART because he was a taxi driver and would not be able to drive while feeling dizzy. A female participant decided not to start ART for fear of falling ill, which might compromise her ability to take care of her ailing husband.

> "After I took the first doses [during the first 2 weeks], my health started changing, my body turned into a very dark color, I was feeling dizzy, [and had] bad dreams. I started having bad dreams [more] than before I started taking this medication. So, I just noticed this medicine was not working properly in me, because I used to sleep well, [and have] a lot of dizziness because of the same medicine and that's how I decided to stop taking it.
>
> –Single man, 33 years old

### Late linkage

Ten participants linked to care, but later than their appointed date and later than the 7-day window recommended by the World Health Organization (WHO) for rapid ART initiation. They reported having to overcome several psychological, social, and structural barriers before returning to the clinic to link to care. Psychologically, participants cited barriers such as denial of their positive HIV test results, perceived and internalized stigma, fear of medication side effects, and a lack of trust in health care workers. As with other participants, one pregnant participant shared feelings of rejection by her sexual partner as a contributing reason for late linkage to care (Box 2). Another participant said that a lack of social support meant that she had no one to encourage her or talk to her about returning to the clinic after her HIV diagnosis, while another participant fell ill and was too weak to go to the clinic. Health system and structural factors also featured prominently. Several participants cited long wait times at the clinic and inflexible work hours contributing to their delay in linking to care.

> "They [health workers] just said we are going to give you ARVs, go back for antenatal [care] and come and collect them later. 'Wait for us, we will give you [ARVs] after we are done [with you].' When I went back for antenatal [care], they told me, 'You'll get your medicine at the end when we are done, between 1200–1300 hours'. We were just seated there waiting for them, and when it reached 1500 hours, I was hungry, so I went home.–Single woman, 25 years old

> **Box 2. Narrative 2: Excerpt showing participant dealing with feelings of rejection and how she overcame it (Single woman, pregnant, 28years old)**
>
> **Participant:** When I got pregnant, I discovered that my man had impregnated another woman before me and moved in with her . . . He was paying more attention to the other woman and he abandoned me and didn't provide anything for me. . .**Interviewer:** What happened in your life or at the clinic that made you go back?**Participant:** . . . I just remembered what we were taught at the clinic about the virus multiplying when someone delays starting treatment and so I convinced myself that I am not the only one who is infected and that I should go and start treatment because this wasn't my fault, I was not promiscuous, but that man brought this disease to me.
>
> Asked what helped them overcome these barriers to ultimately link to care, albeit late, participants mentioned continued follow-up by clinic staff (via phone call or home visit), prolonged patient-centered counselling and engagement with clinic staff, and disclosure to family and friends who encouraged and facilitated linkage to care through counselling, reminders, or physically picking up ART on their behalf. They also drew encouragement from others who were already on treatment and healthy. A few fell seriously ill and had no choice but to return to the clinic to link to care and treatment.

> "After I saw how those who are infected suffer, I got scared. I started asking myself, 'So even I will get sick to the point of being carried on a stretcher?' No, let me just start on my own while I am still energetic."–Single woman, 25 years old

> "After the people from church, the elders my mother invited, and my mother talked to me, I heard what they said. The doctor also followed me home and told me that I should not

choose to ruin my life and advised me to start treatment. He even brought me Soya porridge because I had become too thin. After that I just decided to start, so I was told to go the following morning and I was there by 06:00 hours"–Singe woman, 30 years old.

### Early successful linkage

The ten participants who successfully linked to care within 90 days of diagnosis said they did so because they were afraid of becoming ill as a result of delayed treatment, were encouraged by their family, received patient-centered care from peers or clinic staff, and/or were motivated to stay healthy to be able to care for their children. The fear of becoming ill was also compounded by the shame of being perceived as sickly, hence the urgency to start and continue ART.

"When one neglects themselves that's where you find that. . . You are being carried in a wheelbarrow like that and a lot of people now get to know . . . [your HIV status]"–Married woman, 23 years old

Though linking early, some participants described needing to overcome feelings of denial and depression after learning of their HIV status, especially those who were not mentally prepared to test when asked to. A participant who learned of her status from the antenatal clinic reported she struggled to accept her status at first and refused to take her medication. However, she was helped by the counsellors at the clinic after her aunt told them about her struggles, and she said her motivation to start ART was to protect her unborn baby. Another male participant felt confused, depressed, and lost his appetite after learning his HIV status.

"At that time, they had told me that they diagnosed me with HIV and that's how I just dashed out [of the clinic]. I felt so confused such that I could not think properly and when I reached home, I couldn't even eat. I went to my bedroom sat down and started praying, asking God what was happening in my life and why I could not eat."–Married man, 24 years old

### Silent transfer

Eleven participants silently transferred to another facility without notifying clinic staff. Of these, eight were women (21% of women) and three were men (15% of men). All self-reported accessing a different health facility from their original clinic to either re-test or collect medication. Five of ten participants opted to attend a clinic that was closer to their home than the original clinic where they were diagnosed with HIV. One of the five also mentioned having greater confidence in the healthcare provider's ability to treat them at the new clinic. The accounts that follow illustrate how participants exhibited agency over their own HIV care and self-efficacy in transferring silently to new clinics that they perceived as offering more convenience, more client-centered services, or a combination of both. In some cases, participants described deliberately bi-passing existing clinic transfer systems by presenting themselves as being unaware of their HIV status at the new clinic of their choice.

"My old clinic is far and this [new] clinic is near home . . . And this clinic is bigger and has doctors who are well qualified. I have also noticed that there are many [non-governmental] organizations working here [at the new clinic] but don't work there [at the old clinic].

These organizations work to ensure that our lives are prolonged."–Married man, 55 years old

Three participants mentioned silently transferring from the clinic where they initially tested due to a lack of privacy and confidentiality, which put them at risk of being stigmatized by those who might see them getting HIV services. One participant alluded to their lack of trust in the health care workers and their ability to maintain confidentiality, which led them to seek services from another clinic.

"My husband did not believe the first diagnosis and so we went to another clinic to re-test . . . And the health workers at this [first] clinic don't keep a secret."–Married woman, 28 years old.

One female participant knew of her HIV status but did not know how to disclose her status to her partner because she found it difficult to discuss with her partner directly. So, she chose to receive couples' counseling and testing at another clinic, under the guise of not knowing her HIV status, as a way of facilitating her status disclosure.

"I found a partner to marry me . . . But found it very difficult to tell him my status. I asked him if we can both do an HIV test and so we came to this clinic where we were both found positive"–Married woman, 34 years old

Another participant narrated that she was poorly treated by the first counselor attending to her, which led her to present to another clinic under the guise of being a new client. She explained that at the first clinic, she was denied a second test to confirm her result and was spoken to rudely.

"After we got tested, they told us the results and then they called my husband's relatives and started telling them the news. So, I asked them how they expected me to get drugs after they have told other people about my status. And that's how they just started talking to me rudely saying, 'If you don't want [to], don't get the drugs . . ., you will come and get them when you get very sick.' I told them 'I wasn't coming back.'"–Married woman 40 years old

## Discussion

In this qualitative study, we present narrative data examining reasons for timely, late, and missed linkage to care in routine care settings in the test and treat era according to a framework conceptualizing linkage to care as its own nuanced pathway instead of a simple one-off step. Within this framework, we describe the experiences of PLHIV in a large HIV treatment program who never linked to care or silently transferred care, which are underreported in the literature due to difficulties in reliably identifying these populations. We observed more men than women who dropped off the linkage pathway early, without enrolling in the national program or collecting ART, and more women who successfully collected ART early, but never returned for a follow up visit or silently transferred their care to a new clinic. In our qualitative data, we detected well-known themes of internal and external stigma, concerns about ART side effects, and a perceived lack of partner and social support—all established in the HIV literature—that impeded linkage to care at every step of the linkage pathway. However, we also uncovered newer themes specific to the current test and treat era related to the rapidity of ART initiation and insufficient patient-centered post-test counseling that each appeared to

exacerbate well-known barriers. Counteracting these factors, we noted patient-centered health worker efforts, and family, peer, and other social network supports that all supported linkage to care. These findings have implications for how to strengthen the linkage pathway and minimize silent transfer and early treatment interruption for large-scale HIV treatment programs in the modern treat all era.

Critically, our findings highlight that, despite recommendations to start all patients on ART immediately, some patients need more time and individualized support to internalize both a new HIV diagnosis and the need to start lifelong medication than the modern test and treat approach may routinely afford. While both PITC and index testing initiatives have led to an increase in testing uptake in Zambia [16, 17], the post-test counselling that accompanies these HTS modalities may be insufficient to meet the individualized needs of patients who are wholly unprepared for a new HIV diagnosis or have major multi-level barriers to starting and staying in HIV care. This need has been noted previously in a study comparing the effectiveness of PITC versus VCT on treatment initiation in Zambia, which found that clients testing through PITC were less likely to start and sustain treatment compared to those voluntarily seeking testing [18, 19]. Our findings suggest that PITC leaves some clients feeling 'forced' to test despite not being psychologically prepared to know their HIV status, and this was especially the case for those who felt that they were not at personal risk for HIV acquisition or had no HIV symptoms. Several participants also described being in a state of psychological "shock" at the time of diagnosis, and were not offered sufficient tailored support or time to think through the implications of their new diagnosis.

Unfortunately, post-test HIV counselling and support in Zambia remains largely a one-size fits all approach, which too often overlooks important psychological and social barriers that prevent clients from experiencing timely linkage to care. For example, the client presented in the case report who did not pick up his ART regimen for two weeks following his diagnosis suffered psychologically and emotionally at learning his results while trying to cope with a broken marriage. Such clients require immediate support from the time of diagnosis, ideally from a healthcare worker who, firstly, can identify that the client is at risk of not linking due to these kinds of psychosocial barriers and, secondly, is skilled in the proper counseling techniques to meaningfully help the client. Systematic approaches to identify and help those most in need of individualized linkage support following HIV diagnosis remain largely elusive. To assist with identifying these clients, screening tools, validated for use in low and middle-income countries, can be used in post-test counseling settings to better engage clients with high internalized stigma, comorbid mental health disorders, or other conditions that may undermine linkage [20]. Once identified, such clients may benefit from differentiated counseling approaches that provide individualized, longitudinal psychological support to help patients navigate through feelings of denial, blame, and perceived stigma, and address mental health barriers such as depression and substance use, to help them take decisive action to start and sustain treatment. Such individualized support can apply evidence-based approaches, such as motivational interviewing techniques that have been shown to improve feelings of self-efficacy, readiness, and motivation for change, as well as decrease feelings of anxiety and depression, which have been associated with improved HIV treatment adherence and could result in similar improvements for achieving full linkage [21, 22]. Other approaches that can be differentiated to support linkage include collaboration between counselor and patient, empathetic listening, anxiety management, behavioral activation, and cognitive coping [23]. Existing differentiated service delivery (DSD) models, such as community adherence groups that currently enable stable HIV clients to meet regularly outside of the clinic for drug refills, symptom screening and psychosocial support, could be adapted to improve linkage support for clients unwilling to access services in clinics [24–26]. While potentially resource intensive in the short-term, these kinds of

tailored interventions can be adapted to support linkage to HIV care, and may become less resource intensive with time as patients achieve viral suppression and require less contacts with the health system to sustain ART.

Mental health disorders may play a particularly prominent, and currently underappreciated, role in delaying linkage given their high prevalence among PLHIV and the lack of integrated services to address these co-morbidities [27–31]. Our findings suggest that the health system and routine HIV treatment program in Zambia are ill equipped to appropriately identify and support clients with possible mental health disorders at the time of HIV diagnosis. That said, a number of emerging individualized therapy approaches, such as cognitive behavioral therapy (CBT) are increasingly being adapted for use in resource-limited settings. In Zambia, for example, the Common Elements Treatment Approach (CETA), a CBT-based trans-diagnostic intervention designed specifically for use in low- and middle-income countries, has demonstrated a significant reduction in depression and trauma among PLHIV with unhealthy alcohol use [23, 32–34].

Experience with, and fears of, ART-associated side effects, including vivid dreams, dizziness, difficulties swallowing pills and changes in skin color have long been reported to negatively impact ART adherence [35, 36], and hindered linkage to care in our study. Community sensitization, quality post-test and medical counseling, health worker training, and the successful transition in most settings to newer dolutegravir-based ART regimens that abandon non-nucleoside reverse transcriptase inhibitors like efavirenz all may help address this barrier. Similarly, reported instances of internalized and externalized HIV-associated stigma are not new, and are consistent with other studies conducted in Zambia and globally highlighting these same challenges [37–40]. While there have been consistent calls for multi-level anti-stigma interventions, including interventions targeted at the social, community, and health system levels and greater education of families, health workers, and community members [40], examples of efficacious interventions with robust evidence showing their effects on changing social perceptions are limited. Such interventions need to address the sociocultural factors that continue to perpetuate negative perceptions of PLHIV. This could be done by changing maladaptive social norms that fuel discrimination and internalized stigma or, by extending and strengthening helpful social networks to promote a more supportive community for PLHIV.

Participants' demonstrated desire and agency to silently transfer to more convenient and patient-centered clinics suggests that current health systems and models of HIV care are not sufficiently structured to ensure privacy, confidentiality, and empathy in service delivery, particularly under the compressed timeline for ART initiation under test and treat policy. Moreover, the health system may pose unnecessary complexity or lack of flexibility in the way it meets clients' psychosocial or clinical needs or responds to clients' concerns around healthcare quality or privacy. The phenomenon of silent transfer may be a direct response to these issues, is becoming increasingly prevalent in the test and treat era given the near ubiquity of testing and treatment venues [41], and may pose hidden burdens to patients and the health system. For patients, the process of secretly transferring from one clinic to another ultimately results in patients needing to submit to unnecessary additional physical examinations, blood draws and counseling sessions while being treated as a "new patient," all of which incur both direct and opportunity costs to patients. For the health system, silent transfer places additional strain on limited healthcare resources, incorrectly classifies patients as experiencing treatment interruption or care disengagement and contributes toward an inaccurate understanding of true HIV care retention and number of PLHIV newly starting ART.

Insights gleaned from our participants about the silent transfer phenomenon in Zambia suggest possible actions that can be taken to re-organize patient services from testing through "full linkage" to streamline the linkage pathway and to re-focus the health system and health

care workers on providing client-centered and patient-friendly HIV services. First, it is imperative that health systems develop patient-centered "first contact" linkage encounters at the diagnosing facility, where patients are made to feel welcomed and accepted to minimize the chances of silent transfer. When patients do express a preference for care transfer, new mechanisms are needed that simplify the process and make the patient feel comfortable revealing a preference for transfer to a new facility to establish or continue HIV care. Second, a focus on patient-centeredness should extend to the receiving clinic, where patients should be made to feel comfortable revealing that they were receiving care elsewhere so that documentation of their past medical history is cohesive and complete. Training of frontline health workers in "customer service" skills can help in this regard.

To put such actions into routine practice will require leveraging ongoing investments in differentiated service delivery accelerated recently by the COVID-19 pandemic. The introduction of test and treat policy combined with a surge in HIV case-finding activities have placed increased burden on healthcare workers, which in turn may adversely affect the time available and willingness of providers to deliver patient-centered services in pursuit of full linkage [42]. External public health shocks posed by the COVID-19 pandemic have complicated matters by creating concerns about SARS-COV-2 transmission in health facilities and disrupting information dissemination about ART service availability [10]. Yet COVID-19 has increased recognition of the importance of more differentiated and community-oriented service delivery systems for PLHIV [10]. Adapting existing and emerging DSD models to extend beyond clients already engaged in HIV care to support new ones may improve linkage to care along the entire pathway, without placing additional burden on healthcare workers. In our study, participants expressed a preference for clinics known to receive NGO support, particularly for how they support the quality, timeliness, and patient-centeredness of HIV testing and clinical services. Thus, NGOs can play a particularly important role in facilitating differentiated services for testing and linkage and can be essential allies in efforts to enhance the quality of services to support robust full linkage.

## Conclusion

Despite greater and simplified access to ART in the modern test and treat era, PLHIV may nonetheless link to care in a non-linear or interrupted fashion and may face multi-level barriers at each step from HIV diagnosis to follow up contact with the health system, resulting in a potentially complex and prolonged process to achieve full linkage to care. PLHIV struggle emotionally, socially, and/or psychologically with a new HIV diagnosis, but often go without the requisite supports needed to process their diagnosis prior to rapid treatment initiation and continue along the pathway to first follow up visit to confirm full linkage to care. It is critical, then, that health systems are re-organized and adapted to identify such patients and provide them with differentiated care promptly upon HIV diagnosis, while improving the competencies and patient-centeredness of health care workers and the settings in which they operate, to ensure successful full linkage to routine HIV care in settings heavily burdened by HIV.

## Supporting information

**S1 Checklist. COREQ (COnsolidated criteria for REporting Qualitative research) checklist.**
(PDF)

## Acknowledgments

We acknowledge the Research Coordinators namely Tikulirekuti Banda, Mwila Lundamo and Ntenje Katota who coordinated field activities and worked closely with the research assistants, guided the recruitment of participants and contributed to the data collection. We also acknowledge Mary Mulungu, Nelly Zulu, Lloyd Chifunda, Kabwe Mwamba, Mabuchi Banda, Besa Chibwe and Mwati Chipungu who were responsible for recruiting and collecting data from participants. We further would like to acknowledge Chisha Nakazwe, Harriet Mugeni and Mwila Yalobi who translated and transcribed audios into verbatim. We thank the staff at the 10 government health facilities who welcomed the data collection team and supported the recruitment process. Finally, we thank all participants for their voluntary participation.

## Author Contributions

**Conceptualization:** Jenala Chipungu, Helene Smith, Michael E. Herce.

**Data curation:** Jenala Chipungu, Chanda Mwamba, Mwiza Haambokoma.

**Formal analysis:** Jenala Chipungu, Chanda Mwamba, Mwiza Haambokoma, Michael E. Herce.

**Funding acquisition:** Theodora Savory, Izukanji Sikazwe, Michael E. Herce.

**Investigation:** Michael E. Herce.

**Methodology:** Jenala Chipungu, Helene Smith, Michael E. Herce.

**Project administration:** Helene Smith, Theodora Savory, Michael E. Herce.

**Supervision:** Theodora Savory, Michael E. Herce.

**Writing – review & editing:** Jenala Chipungu, Helene Smith, Chanda Mwamba, Mwiza Haambokoma, Anjali Sharma, Theodora Savory, Maurice Musheke, Jake Pry, Carolyn Bolton, Izukanji Sikazwe, Michael E. Herce.

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
