## [Decision Letter · Decision Letter 0]

4 Feb 2024

PGPH-D-23-02178

An Exploration of Multi-level Factors Affecting Routine Linkage to HIV Care in Zambia’s PEPFAR-supported Treatment Program

Dear Dr. Chipungu,

Thank you for submitting your manuscript to PLOS Global Public Health. After careful consideration, we feel that it has merit but does not fully meet PLOS Global Public Health’s publication criteria as it currently stands. Therefore, we invite you to submit a revised version of the manuscript that addresses the points raised during the review process.

EDITOR:

While this manuscript reports on important work, the reviewers identified a number of concerns that would need to be addressed before this meets publication criteria.  A more thorough literature review will help you address Reviewer 2's comments, but do not feel obligated to use all the citations they suggest.  I do agree that the use of the COREQ checklist will support more rigorous reporting of your qualitative methods and findings. 

Authors should review the PLOS Global Public Health’s publication criteria to help guide revisions to the manuscript. 

We look forward to receiving your revised manuscript.

Kind regards,

Sarah E. Brewer, PhD

Academic Editor

Journal Requirements:

2. Please provide separate figure files in .tif or .eps format only and remove any figures embedded in your manuscript file. Please also ensure all files are under our size limit of 10MB.

Additional Editor Comments (if provided):

Reviewers' comments:

Reviewer's Responses to Questions

**Comments to the Author**

1. Does this manuscript meet PLOS Global Public Health’s publication criteria? Is the manuscript technically sound, and do the data support the conclusions? The manuscript must describe methodologically and ethically rigorous research with conclusions that are appropriately drawn based on the data presented.

Reviewer #1: Yes

Reviewer #2: Yes

2. Has the statistical analysis been performed appropriately and rigorously?

Reviewer #1: Yes

Reviewer #2: N/A

3. Have the authors made all data underlying the findings in their manuscript fully available (please refer to the Data Availability Statement at the start of the manuscript PDF file)?

Reviewer #1: Yes

Reviewer #2: Yes

4. Is the manuscript presented in an intelligible fashion and written in standard English?

Reviewer #1: Yes

Reviewer #2: Yes

5. Review Comments to the Author

Reviewer #1: The authors present the results of a qualitative study of facilitators and barriers to full linkage to ART from 10 facilities in Lusaka, Zambia. While the paper addresses an important topic and is well written, there are several areas the authors may wish to consider.

Minor comments:

1) please include information on the IRB or ethics committee that approved the study.

2) It is unclear how the authors established "silent transfer" from the routine records. If the definition is transfer without notifying the first clinic it would imply that this was not in the records. Was silent transfer status established after contacting the potential participant?

3) figure 1 either doesn't need the single arrow from reasons to VCT or arrows from reasons need to be added going to PITC and CBT.

4) no need to abbreviate if only used once (e.g. LMIC on line 437)

Major comments:

1) Table 2 would make more sense as row percentages rather than column percentages. If the authors are looking to interpret the characteristics that are associated with the linkage stage of the participant, some nuances are lost when looking within linkage stage rather than among those with a particular characteristic what is the distribution of linkage stage within that characteristic.

2) while the quotes from the qualitative interviews are good, since the purpose of the paper is to identify barriers and facilitators of full linkage, it may be more helpful to also identify the linkage stage along with marital status, sex, and age in the reasons for testing section.

3) in Table 2 you report 11 people were silent transfers but on line 356-357 you report 10 participants 8 women and 2 men rather than 3 men as shown in table 2.

4)Similarly comparing whole numbers of women to men in the results is not very informative. There are more women in your sample and there are more women living with HIV in Zambia than men. when discussing linkage stage, silent transfer for example, rather than saying 8 and 2 (or 8 and 3 as in table 2) those represent 21% of women and 15% of men. Presenting as frequencies ignores the overall sex imbalance, e.g. you would expect more women by chance alone.

Reviewer #2: Comment to authors

This is a very well-written manuscript.

INTRODUCTION

The 2016 data presented in the first two paragraphs about ART use are very old, authors need to provide the current data.

It is great that this study focuses on a very important aspect for people living with HIV. However, several aspects need to be addressed to improve the manuscript:

1. The main focus of this paper is on understating enablers and barriers to the linkage to HIV care, but there is no real synthesis of findings of the existing studies globally and in Zambia about (i) enablers/facilitators to the linkage to HIV care/service, and (ii) barriers to the linkage to HIV care/services.

2. This synthesis will help you to identify research/knowledge gap(s) in the literature, which you bridge through your study, and that becomes your contribution to knowledge. In other words, what is new (novelty) about your study? Why is it important to research this particular topic with this study participants?

I would suggest the authors go through the following studies and look at the references used to help you make the synthesis. This synthesis will be very helpful once you discuss the findings.

Stigma and Discrimination towards People Living with HIV in the Context of Families, Communities, and Healthcare Settings: A Qualitative Study. Int. J. Environ. Res. Public Health 2021, 18, 5424. https://doi.org/10.3390/ijerph18105424

HIV Stigma and Discrimination: Perspectives and Personal Experiences of Healthcare Providers in Yogyakarta and Belu, Frontiers in Medicine https://www.frontiersin.org/articles/10.3389/fmed.2021.625787/full

Barriers to HIV testing among male clients of female sex workers in Indonesia International Journal for Equity in Health 17 (1), 1-10 https://link.springer.com/article/10.1186/s12939-018-0782-4

Barriers to Accessing HIV Care Services in Host Low and Middle Income Countries: Views and Experiences of Male Ex-Migrant Workers Living with HIV. International Journal of Environmental Research and Public Health 19 (21), 14377. https://www.mdpi.com/1660-4601/19/21/14377

Traditional Human Immunodeficiency Virus treatment and family and social influence as barriers to accessing HIV care services in Belu, Indonesia. PloS one 17 (7), e0264462

https://journals.plos.org/plosone/article?id=10.1371/journal.pone.0264462

Barriers to access to antiretroviral therapy by people living with HIV in an Indonesian remote district during the COVID-19 pandemic: a qualitative study. BMC Infectious Diseases 23 (1), 296

https://link.springer.com/article/10.1186/s12879-023-08221-z

Perceptions among transgender women of factors associated with the access to HIV/AIDS-related health services in Yogyakarta, Indonesia. PloS one 14 (8), e0221013

https://journals.plos.org/plosone/article?id=10.1371/journal.pone.0221013

Facilitators to accessibility of HIV/AIDS-related health services among transgender women living with HIV in Yogyakarta, Indonesia. AIDS research and treatment 2019

https://www.hindawi.com/journals/art/2019/6045726/

Risk factors and The Impact of HIV among Women Living with HIV and Their Families in Yogyakarta and Belu District, Indonesia

Flinders University, College of Medicine and Public Health. (see chapter 7)

https://www.researchgate.net/profile/Nelsensius-Fauk/publication/361886192_Risk_factors_and_the_impact_of_HIV_among_women_living_with_HIV_and_their_families_in_Yogyakarta_and_Belu_district_Indonesia/links/62cab6a400d0b4511046b7a9/Risk-factors-and-the-impact-of-HIV-among-women-living-with-HIV-and-their-families-in-Yogyakarta-and-Belu-district-Indonesia.pdf?utm_source=nationaltribune&utm_medium=nationaltribune&utm_campaign=news

METHODS

Who did the interviews. I would suggest the authors using COREQ checklist to help improve their reporting on the methods section: https://www.equator-network.org/reporting-guidelines/coreq/

Who did the transcription? Who are the 3 coders? Are the coders the interviewers? If they are different people, how did you manage the non-verbal language, gestures, etc that were observed during interviews? How did you solve discrepancies among the 3 coders during the coding process? Authors need to provide the thematic analysis steps used.

DISCUSSION

“Recurrent themes of internal and external stigma, concerns about medication side effects, and a perceived lack of partner and social support, appear to impede linkage to care at every step along the linkage pathway, while quality counseling, patient-centered health worker engagement, and family, peer, and other social network supports seemed to help overcome these barriers.”

These findings are not new and have been reported in previous studies. Please consult the following studies and the references they used:

Stigma and Discrimination towards People Living with HIV in the Context of Families, Communities, and Healthcare Settings: A Qualitative Study. Int. J. Environ. Res. Public Health 2021, 18, 5424. https://doi.org/10.3390/ijerph18105424

HIV Stigma and Discrimination: Perspectives and Personal Experiences of Healthcare Providers in Yogyakarta and Belu, Frontiers in Medicine https://www.frontiersin.org/articles/10.3389/fmed.2021.625787/full

Barriers to Accessing HIV Care Services in Host Low and Middle Income Countries: Views and Experiences of Male Ex-Migrant Workers Living with HIV. International Journal of Environmental Research and Public Health 19 (21), 14377. https://www.mdpi.com/1660-4601/19/21/14377

Perceptions among transgender women of factors associated with the access to HIV/AIDS-related health services in Yogyakarta, Indonesia. PloS one 14 (8), e0221013

https://journals.plos.org/plosone/article?id=10.1371/journal.pone.0221013

Facilitators to accessibility of HIV/AIDS-related health services among transgender women living with HIV in Yogyakarta, Indonesia

AIDS research and treatment 2019

https://www.hindawi.com/journals/art/2019/6045726/

CONCLUSION

What are the implications of your findings for future studies and HIV policies and practice in the study setting?

6. PLOS authors have the option to publish the peer review history of their article (what does this mean?). If published, this will include your full peer review and any attached files.

**Do you want your identity to be public for this peer review?** For information about this choice, including consent withdrawal, please see our Privacy Policy.

Reviewer #1: No

Reviewer #2: No

[NOTE: If reviewer comments were submitted as an attachment fil

---

## [Decision Letter · Decision Letter 1]

25 Apr 2024

An Exploration of Multi-level Factors Affecting Routine Linkage to HIV Care in Zambia’s PEPFAR-supported Treatment Program in the Treat All Era

PGPH-D-23-02178R1

Dear Miss Chipungu,

We are pleased to inform you that your manuscript 'An Exploration of Multi-level Factors Affecting Routine Linkage to HIV Care in Zambia’s PEPFAR-supported Treatment Program in the Treat All Era' has been provisionally accepted for publication in PLOS Global Public Health.

Best regards,

Sarah E. Brewer, PhD

Academic Editor

Reviewer Comments (if any, and for reference):

Reviewer's Responses to Questions

**Comments to the Author**

1. If the authors have adequately addressed your comments raised in a previous round of review and you feel that this manuscript is now acceptable for publication, you may indicate that here to bypass the “Comments to the Author” section, enter your conflict of interest statement in the “Confidential to Editor” section, and submit your "Accept" recommendation.

Reviewer #2: All comments have been addressed

2. Does this manuscript meet PLOS Global Public Health’s publication criteria? Is the manuscript technically sound, and do the data support the conclusions? The manuscript must describe methodologically and ethically rigorous research with conclusions that are appropriately drawn based on the data presented.

Reviewer #2: Yes

3. Has the statistical analysis been performed appropriately and rigorously?

Reviewer #2: N/A

4. Have the authors made all data underlying the findings in their manuscript fully available (please refer to the Data Availability Statement at the start of the manuscript PDF file)?

Reviewer #2: (No Response)

5. Is the manuscript presented in an intelligible fashion and written in standard English?

Reviewer #2: Yes

6. Review Comments to the Author

Reviewer #2: Thanks to the authors who have improved the manuscript. All my comments have been sufficiently addressed. I have no further comments.

7. PLOS authors have the option to publish the peer review history of their article (what does this mean?). If published, this will include your full peer review and any attached files.

**Do you want your identity to be public for this peer review?** For information about this choice, including consent withdrawal, please see our Privacy Policy.

Reviewer #2: No
